# Dietary Silk Peptide Prevents High-Fat Diet-Induced Obesity and Promotes Adipose Browning by Activating AMP-Activated Protein Kinase in Mice

**DOI:** 10.3390/nu12010201

**Published:** 2020-01-13

**Authors:** Kippeum Lee, Heegu Jin, Sungwoo Chei, Jeong-Yong Lee, Hyun-Ji Oh, Boo-Yong Lee

**Affiliations:** 1Department of Food Science and Biotechnology, College of Life Science, CHA University, Seongnam 13488, Korea; 2Worldway Co., Ltd., Sanda-gil, Jeonul-myeon, Sejong-si 30003, Korea

**Keywords:** AMP-activated protein kinase, browning, fatty acid oxidation, obesity, silk peptide

## Abstract

Obesity is associated with metabolic syndrome and other chronic diseases, and is caused when the energy intake is greater than the energy expenditure. We aimed to determine the mechanism whereby acid-hydrolyzed silk peptide (SP) prevents high-fat diet-induced obesity, and whether it induces browning and fatty acid oxidation (FAO) in white adipose tissue (WAT), using in vivo and ex vivo approaches. We determined the effects of dietary SP in high-fat diet-fed obese mice. The expression of adipose tissue-specific genes was quantified by western blotting, qRT-PCR, and immunofluorescence analysis. We also investigated whether SP directly induces browning in primarily subcutaneous WAT-derived adipocytes. Our findings demonstrate that SP has a browning effect in WAT by upregulating AMP-activated Protein Kinase (AMPK) phosphorylation and uncoupling protein 1 (UCP1) expression. SP also suppresses adipogenesis and promotes FAO, implying that it may have potential as an anti-obesity drug.

## 1. Introduction

Obesity is defined as a state of excessive adipose tissue expansion arising from an imbalance between energy intake and expenditure [1,2]. Due to economic development and changes in diet and lifestyle, obesity has become a global epidemic. Obesity is regarded as a major health problem that predisposes toward type 2 diabetes, cardiovascular disease, hypertension, stroke, atherosclerosis, non-alcoholic fatty liver disease, and various cancers [3,4].

In mammals, energy balance involves two types of adipose tissue: white adipose tissue (WAT), which stores energy in the form of triglycerides and can expand through changes in cell size (hypertrophy) and cell number (hyperplasia) [5,6], and brown adipose tissue (BAT), which dissipates stored energy as heat through the expression of uncoupling protein 1 (UCP1) [7,8,9]. UCP1 promotes proton leakage in the mitochondrial membrane, thereby dissipating the energy released by oxidation instead of using it for ATP synthesis. Recently, it has been shown that there is also a third type of adipocyte, in which UCP1 expression can be induced by cold exposure or β3-adrenoceptor agonists, and which is referred to as a beige or BAT-like adipocyte [10].

An increase in WAT-to-BAT transdifferentiation is an important mechanism that is used to combat fat accumulation. This “browning” involves mitochondrial expansion and fatty acid oxidation (FAO), which increase energy expenditure. Fatty acids are required for the UCP1-mediated uncoupling mechanism, and numerous studies have shown that cold stimulates FAO in mice, and this is required for thermogenesis [11,12,13]. Carnitine palmitoyltransferase 1 (CPT1), which is an enzyme required for the transport of long-chain fatty acids from the cytoplasm into the mitochondria, is a rate-limiting factor in FAO [14]. In addition, the browning process requires high levels of expression of PR domain-containing 16 (PRDM16) and peroxisome proliferator-activated receptor-gamma co-activator 1α (PGC1α) as regulators. Both these transcription factors promote the expression of mitochondrial genes, including UCP1, and mitochondrial biogenesis [15,16,17]. Therefore, the identification of a nutritional activator of UCP1 expression in WAT may lead to the development of a new class of therapeutic agents for obesity [18].

Acid-hydrolyzed silk peptide (SP), derived from *Bombyx mori* cocoons, is a dietary biomaterial that has various applications in biotechnology, bio-pharmacology, cosmetic, and food industries in Asian countries [19]. It is known to be biocompatible and its administration has not been associated with any documented side effects. Recently, in vitro and in vivo studies have shown that SP has beneficial effects on metabolism and health [20,21]. For instance, it has been reported that treatment with silk fibroin enhances insulin sensitivity and glucose uptake in 3T3-L1 adipocytes and type 2 diabetic mice [22,23]. In addition, silk fibroin proteins have been shown to attenuate adipogenesis in adipocytes and C57BL/6N mice [24,25], and to increase fat oxidation in exercising mice [26,27]. However, the molecular mechanisms whereby SP may induce browning and fatty acid oxidation in WAT have yet to be established. Therefore, we aimed to determine the effects of dietary SP on the metabolism of high-fat diet (HFD)-induced obese mice and in subcutaneous white adipose tissue (sWAT)-derived primary cells.

## 2. Materials and Methods

### 2.1. Preparation of SP from Bombyx Mori

Dietary SP (lot number, 1803002) was prepared from the cocoons of *Bombyx mori* and was obtained from Worldway Co., Ltd. (Sejong, Korea). As shown in Figure 1, raw cocoons were acid-hydrolyzed, and the resulting solution was neutralized, decolorized, filtered, desalted, and freeze-dried to obtain a pale yellow powder. The nutrient composition of the SP was analyzed by International official methods of analysis (AOAC) methods, and the results are presented in Table 1. In detail, carbohydrates and sugars in an activated charcoal column and a later elution with different proportions of ethanol (AOAC 954.11) to fractionate them selectively according to their degree of polymerization. Crude fat of SP were extracted using the soxhlet apparatus with hexane for 4 h (AOAC 920.153) and determined gravimetrically. Crude protein was estimated by AOAC 968.06 method, through an acid digestion and nitrogen distillation using Kjeldahl method. Lastly, sodium content in SP was conducted following the AOAC 984.27 method. The mean molecular weights of SP were measured by MicroQ-TOF III mass spectrometry (Bruker Daltonics, Hamburg, Germany). The SP sample which is dissolved in 10 mM Sodium phosphate buffer with methanol (4:1) was then injected into an UltiMate 3000 high-performance liquid chromatography (HPLC) system (Dionex, Sunnyvale, CA, USA), and a Poroshell 120 EC-C18 column (2.1 mm × 100 mm, 2.7 μm) was used to analyze. Acetonitrile containing 0.2% formic acid and 0.2% formic acid in water were used as the mobile phases (at ratios of 95:5 to 5:95 (*v/v*), with a flow rate of 0.2 mL/min), and the results are presented in Figure 2. Lastly, the free amino acid composition of SP was determined using a 2695 HPLC system (Waters, Milford, MA, USA). An AccQ-Tag Amino acid analysis column (silica C_18_, 3.9 mm × 150 mm) and a Waters 2475 Multiλ Fluorescence detector were used for the SP analysis, which is described in Figure 3.

### 2.2. Cell Culture and Treatment

The stromal vascular fraction was obtained from the sWAT of 6–8-week-old male institute of cancer research (ICR) mice. As described previously [28], isolated sWAT pads were minced and digested in phosphate-buffered saline (PBS) containing 1.5 U/mL of collagenase D (Roche Diagnostics GmbH, Mannheim, Germany), 2.4 U/mL of Dispase II (Roche, Basel, Switzerland), and 10 mM CaCl_2_ for 1 h. The digest was then filtered through 40 μm cell strainers (SPL Life Science, Gyeonggi-do, Korea) three times, washed in PBS, and centrifuged at 1000× *g* for 10 min.

Glutamax DMEM/F12 medium containing 10% FBS and 1% penicillin/streptomycin solution (*P/S*) were used for the maintenance of primary sWAT cells. Two days after the cells reached confluence, this medium was replaced by differentiation medium (DMEM supplemented with 10% FBS, 1% *P/S*, 100 μM indomethacin, 0.5 mM 3-isobutyl-1-methylxanthine, 1 μM dexamethasone, and 5 μg/mL insulin) for two days, and subsequently, the differentiated adipocytes were maintained in DMEM containing 10% FBS, 1% *P/S*, and 5 μg/mL insulin.

SP was prepared as a 400 mM stock solution in distilled water and then diluted with medium to 25, 50, 100, 200, or 400 μM. To induce browning in primary sWAT-derived adipocytes, the cells were incubated in differentiation medium containing 10 nM triiodothytonine (T_3_) and 1 μM rosiglitazone (Rsg). The compounds 6-[4-(2-piperidin-1-ylethoxy)phenyl]-3-pyridin-4-ylpyrazolo[1,5-]pyrimidine (dorsomorphin) (5 μM) or 5-aminoimidazole-4-carboxamide ribonucleotide (AICAR) was added to the differentiation medium to determine the role of AMPK.

### 2.3. Animals and Diet

Four-week-old male ICR mice were purchased from Joong-Ah Bio (Suwon, Korea). The animal experiments and care were approved by the Institutional Animal Care and Use Committee (IACUC) of CHA University (IACUC approval number, 180161). Mice were housed for 1 week under a 12 h light/dark cycle for adaptation. Mice were randomly allocated to four groups (*n* = 10 per group). The first group was fed a chow diet (CD, 10% of calories derived from fat, D12450B, Research Diets, New Brunswick, NJ, USA), the second group was fed an HFD (60% of calories derived from fat, D12492, Research Diets), the third group was fed an HFD and administered 50 mg/kg/day SP (HFD + SP50), and the final group was fed an HFD and administered 200 mg/kg/day SP (HFD + SP200). Each SP concentration was derived from the human doses (0.25 g/60 kg/day and 1 g/60 kg/day) in mathematical table, as described [29]. SP was orally administered to the mice by gavage daily for 6 weeks. The body mass, food intake, and water consumption of the mice were measured weekly. At the end of the treatment period, the mice were euthanized, and their tissues were collected for analysis and weighed.

### 2.4. Rectal Temperature Measurement

At the end of the 6 week treatment period, the rectal temperatures of the mice were measured three times using a Testo 925 Type Thermometer (Testo, Lenzkirch, Germany).

### 2.5. Serum Biochemistry

After the treatment, the mice were fasted overnight and blood samples were collected by cardiac puncture. Serum samples were separated by centrifugation at 4 °C and 4000× *g* for 10 min after the blood had clotted. Commercial enzyme-linked immunosorbent assay (ELISA)/calorimetric assay kits (Abcam and Biocompare, Burlingame, CA, USA) were used to measure serum triglyceride (ab65336), total cholesterol (ab65390), high-density lipoprotein (HDL)-cholesterol (EKC37055), low-density lipoprotein (LDL)-cholesterol (EKC41016), leptin (ab100718), alanine aminotransferase (ALT, ab105134), aspartate aminotransferase (AST, ab133878), and creatinine (ab65340) concentrations. Absorbances were measured at appropriate wavelengths using a plate reader (BioTek Instruments Inc. Winooski, VT, USA).

### 2.6. Histological Analysis and Immunofluorescence Staining

After euthanasia, sWAT and visceral WAT (vWAT) were rapidly collected and fixed in 4% paraformaldehyde solution for 48 h. The tissues were then paraffin-embedded and the resulting blocks were cut into 5–10 µm sections and stained with hematoxylin and eosin (H&E) to assess adipose histology. Photomicrographs were obtained using a Nikon Eclipse E600 microscope (Nikon Corporation, Tokyo, Japan).

For immunofluorescence analysis, fixed cells and WAT sections were stained with rabbit anti-UCP1 (dilution, 1:200) or anti-CPT1 (dilution, 1:1000) antibodies overnight at 4 °C in a moist chamber. Fluorescein isothiocyanate (FITC)-conjugated (dilution, 1:1000) and Alexa Fluor™ 594-conjugated (dilution, 1:1000) secondary antibodies were used. Mitochondria were identified by staining with 1 mM MitoTracker Red (Cell Signaling Technology, Danvers, MA, USA), according to the manufacturer’s protocol, and nuclei were stained using DAPI (Thermo Fisher Scientific, Waltham, MA, USA) fluorescence. After mounting using ProLong Gold Antifade reagent (Thermo Fisher Scientific), fluorescence images were captured using a Zeiss confocal laser scanning microscope (LSM880; Carl Zeiss, Oberkochen, Germany) and Zen 2012 software (Carl Zeiss).

### 2.7. Oil Red O Staining

After 8–10 days of differentiation of the primary cells, they were fixed in 10% formaldehyde for 1 h at room temperature and then washed twice with PBS. The fixed cells were stained with Oil red O (ORO) solution in 6:4 (*v/v*) isopropanol:distilled water for 30 min. After rinsing and drying, the stained adipocytes were imaged and the stain was eluted using 100% isopropanol. The absorbances of these eluates were then measured at 490 nm. Cryostat sections (5 µm) of liver were also stained with 0.1% (*w/v*) ORO solution to visualize hepatic lipid accumulation.

### 2.8. Western Blot Analysis

Cells or tissue was lysed in lysis buffer (iNtRON Biotechnology, Seoul, Korea) supplemented with phosphatase and protease inhibitors. The total protein contents of the lysates were quantified using a protein assay kit (Bio-Rad, Hercules, CA, USA) and then equalized. Samples were separated by SDS-PAGE and transferred to nitrocellulose membranes. These membranes were blocked using 5% non-fat dried milk for 1 h and then incubated overnight at 4 °C with primary antibodies targeting AMPKα, phospho-AMPKα (Thr172), phosphor-hormone-sensitive lipase (p-HSL, Ser563), adipose triglyceride lipase (ATGL, Cell Signaling Technology), fatty acid-binding protein 4 (FABP4), CCAAT enhancer-binding protein α (C/EBPα), diacylglycerol acyltransferase 1 (DGAT1), glyceraldehyde 3-phosphate dehydrogenase (GAPDH), PGC1α, peroxisome proliferator-activated receptor alpha (PPARα), PRDM16 (Santa Cruz Biotechnology, Santa Cruz, CA, USA), sterol regulatory element-binding transcription factor 1 (SREBP1), CPT1, or UCP1 (Abcam, Cambridge, UK). After washing, the membranes were incubated for 4 h with secondary antibodies conjugated with horseradish peroxidase (1:1000, Santa Cruz Biotechnology) in 5% non-fat dried milk. Reactive band was obtained by chemiluminescence and LAS image software (Fuji, Valhalla, NY, USA).

### 2.9. Quantitative Reverse Transcription Polymerase Chain Reaction Analysis

RNA was isolated from adipocytes or homogenized tissues using TRIzol reagent (Invitrogen, Carlsbad, CA, USA). cDNA was then obtained from 1 μg RNA on a thermal cycler (Bio-Rad) using a Maxime RT PreMix (iNtRON Biotechnology, Seongnam, Korea) for 60 min. The cDNA was amplified using SYBR Green (Roche Diagnostics GmbH, Mannheim, Germany) and a Bio-Rad CFX96 Real-Time Detection System (Bio-Rad). Expression data were normalized to 18S. The mRNA levels were calculated as a ratio, using the 2^−ΔΔ*C*T^ method by using Bio-Rad software (Quantity One 4.62; Bio-Rad), for comparing the relative mRNA expression levels between different groups in the qPCR. The sequences of the primers used in this study are listed in Table 2.

### 2.10. Statistical Analysis

Protein expression data and relative cell diameter data are summarized as mean ± SD and were analyzed using one-way ANOVA and Duncan’s test (SPSS, Chicago, IL, USA). Values with different letters are significantly different; *p* < 0.05 (a > b > c > d > e). Tissue masses and qPCR data are expressed as mean ± SD, and Student’s *t*-test was used to analyze the data. Organ mass and mRNA data were analyzed using Student’s *t*-test, and values were considered significant when * *p* < 0.05 and ** *p* < 0.01.

## 3. Results

### 3.1. Nutritional Analysis of the Silk Peptide

The nutritional components of the SP preparation are shown in Table 1. The dietary SP contained a high concentration of protein (86.80 g/100 g) and low concentrations of carbohydrates (6.79 g/100 g), sugar (0.94 g/100 g), and fat (0.01 g/100 g) on a dry-matter basis. The mass spectrometry analysis (Figure 2) showed that the mean molecular weight of the components of the SP ranged from 150 to 300 Da and that of the SP was <500 Da. Further quantitative analysis of the SP preparation was performed by HPLC and the data are presented in Figure 3. The concentrations of each free amino acid were calculated from the experimental peak area by interpolation using standard calibration curves and L-2-aminobutyric acid as a standard. The protein component of the SP was mainly composed of glycine (33.06%), alanine (28.09%), and serine (11.09%), and contained smaller amounts of other amino acids: valine (2.67%), tyrosine (2.46%), aspartic acid (2.45%), glutamic acid (1.78%), threonine (1.22%), cysteine (1.04%), isoleucine (0.76%), proline (0.74%), leucine (0.72%), arginine (0.50%), phenylalanine (0.44%), lysine (0.38%), histidine (0.38%), and methionine (0.08%), on a dry-matter basis, as shown in Table 3.

### 3.2. SP Inhibits Body Mass Gain in HFD-Induced Obese Mice

To investigate the effects of SP on obesity in vivo, we administered 50 mg/kg/day or 200 mg/kg/day to HFD-induced obese mice. The mice were fed a CD or an HFD and some were treated with SP for 6 weeks.

The mean body mass of the HFD-fed mice was much higher than that of the CD-fed mice, but the body mass of the treatment groups was dose-dependently reduced by SP administration. The HFD-fed mice gained a mean of 24.7 ± 1.2 g, whereas the SP-treated HFD-fed groups (50 or 200 mg/kg/day) gained only 20.5 ± 3.5 g or 11.1 ± 3.4 g, respectively (Figure 4A,B). As shown in Figure 4C,D, the sWAT and vWAT masses were much higher in the HFD group than in the CD group. However, the SP-treated groups had much lower fat pad masses than the HFD control group: the masses of the sWAT and vWAT were 58.8% and 50.0%, respectively, which were lower in the HFD + SP200 group than in the HFD group. However, there was no significant difference in food or water intake among the mouse groups (Figure 4E,F).

There were no significant differences in serum creatinine concentration, a marker of renal injury (Figure 4G), implying that SP treatment did not induce acute kidney injury during the 6 weeks of the study. The serum leptin concentration was 4.0 ± 1.0 ng/mL in the HFD group but only 3.3 ± 1.3 and 2.4 ± 0.6 ng/mL in the SP50 and SP200 groups, respectively (Figure 4H). The triglyceride, total cholesterol, and LDL-cholesterol concentrations were all significantly higher in the HFD group than in the CD group, but SP treatment normalized these concentrations (Table 4). In addition, the HDL-cholesterol concentration was lower in the HFD group than in the CD group, but this defect was ameliorated in the SP-treated groups. Thus, 6 weeks of treatment with SP ameliorates the obesity and dyslipidemia of HFD-fed mice.

### 3.3. SP Reduces WAT Depot Size and Downregulates Adipogenic Gene Expression

To explore the potential anti-obesity mechanisms involved in the effects of SP treatment, the adipocyte size and expression of adipogenic transcription factors were quantified in sWAT and vWAT.

As shown in Figure 5A, adipocyte size was measured in H&E-stained sections. Both depots contained considerably larger adipocytes in the HFD group than in the CD group, but SP treatment reduced the diameter of the adipocytes in a dose-dependent manner, such that the adipocyte size of the sWAT of the HFD + SP200 group was similar to that of the CD group. These results are consistent with the inhibition of body mass gain described above.

The expression of C/EBPα, FABP4, and DGAT1 was also measured using western blotting. These proteins are key mediators of lipid deposition that are induced in the later phases of adipocyte differentiation and are activated in mature adipose tissue. In addition, the inhibition of DGAT1 is known to increase energy expenditure and protect against diet-induced obesity [30]. In the present study, SP treatment reduced the expression of C/EBPα, FABP4, and DGAT1 in a dose-dependent fashion. This is consistent with SP treatment suppressing HFD-induced lipid accumulation by reducing the expression of adipogenic transcription factors in WAT.

### 3.4. SP Induces FAO in WAT

Over the last decade, several studies have shown that an increase in FAO is associated with the activation of mitochondrial biogenesis in BAT [31,32], and these fatty acids are normally generated by lipolysis in adipose tissue [28]. Thus, lipid catabolism is necessary for thermogenesis in BAT or WAT with a BAT-like phenotype. In the present study, we determined the effect of SP on FAO in HFD-induced obese mice by measuring the expression of key mediators using western blotting and qRT-PCR.

As shown in Figure 6, AMPK phosphorylation was much lower in the HFD group than in the CD group, but this effect was dose-dependently ameliorated by SP treatment. AMPK is a vital energy sensor that preserves metabolic homeostasis [33], and the increase in AMPK phosphorylation induced by SP treatment may be responsible for upregulating several pathways, including lipolysis, FAO, and browning. The phosphorylation of HSL, and HSL and ATGL expression, were also increased by the SP treatment of HFD-induced mice. HSL and ATGL are lipolytic enzymes that hydrolyze the ester bonds of triglycerides to release fatty acids and glycerol, and can induce browning in WAT [34]. In addition, the expression of Acox1, Cyp4a10, Cyp4a14, PPARα, and CPT1 were measured. CPT1, which is located on the mitochondrial outer membrane, is an important regulator of mitochondrial FAO, and SP increased the expression of both PPARα and CPT1, which may imply an upregulation of FAO in WAT. In addition, SP treatment increased the mRNA expression of Acox1, Cyp4a10, and Cyp4a14, which are required for peroxisomal and mitochondrial FAO [28]. Taking these findings together, we have shown that SP induces the expression of genes involved in lipolysis and FAO, which may be mediated through the phosphorylation of AMPK.

### 3.5. SP Promotes Browning in WAT

Some WAT is known to have a capacity to modify its metabolic phenotype to one that resembles BAT. This event is referred to as “browning”, and the adipocytes involved are called “beige adipocytes” [6]. Beige cells in WAT have similar characteristics to BAT, including a higher mitochondrial content and the expression of BAT-specific genes, such as PRMD16, PGC1α, and UCP1. To evaluate the effect of SP on browning, the expression of these key BAT markers was quantified (Figure 7A,B). SP treatment increased the expression of PRDM16, PGC1α, UCP1, and UCP3, which are required for thermogenesis, in both WAT depots. The expression of UCP1 in the HFD-fed control group was lower than that in the CD group, but 200 mg/kg/day SP increased this by 1.59-fold and 1.51-fold in the sWAT and vWAT, respectively. In addition, SP treatment markedly increased the mRNA expression of *NRF1* and *TFAM*, which are also important for mitochondrial biogenesis and FAO (Figure 7C,D).

Interestingly, the expression levels of many of these thermogenic markers were increased by SP to a greater extent in sWAT than in vWAT. The mRNA expression of *Prdm16*, which is a master regulator of brown adipocyte differentiation, was increased 4.41-fold in sWAT (*p* < 0.01) and tended to be increased 2.42-fold in vWAT by 200 mg/kg/day SP treatment. SP also increased the mRNA expression of *Pgc1a* 2.80-fold in sWAT (*p* < 0.01) and 2.62-fold in vWAT (*p* < 0.01), and increased the mRNA expression of *Ucp1* 2.66-fold (*p* < 0.05) and 2.28-fold (*p* < 0.05), respectively. These data are consistent with recent studies showing that the browning of sWAT induced by β3 adrenergic agonists or cold exposure is more profound than that of vWAT [35]. Finally, immunostaining showed that the expression of CPT1 and UCP1 in the SP treatment groups was higher than that in the HFD control group (Figure 7E,F). CPT1 expression in WAT is closely related to the induction of UCP1, as part of the upregulation of mitochondrial uncoupling alongside FAO.

BAT-like adipocytes are known to help maintain body temperature through FAO and non-shivering thermogenesis mediated via UCP1 [36]. To determine whether SP treatment affected energy expenditure, rectal body temperature was measured at the end of the treatment period. SP significantly increased rectal temperature to 37.80 ± 0.36 °C and 37.98 ± 0.30 °C at the SP50 and SP200 groups, respectively (Figure 7G). However, there was no statistically significant difference in rectal temperature between the HFD and CD groups (*p* < 0.05). Taken together, SP appears to increase differentiation into BAT-like adipocytes in WAT, demonstrated by increases in UCP1 activation and rectal temperature.

### 3.6. SP Ameliorates Hepatic Lipid Accumulation and Increases FAO in the Liver

HFD-feeding causes hepatic steatosis in mice, which models a stage of non-alcoholic fatty liver disease in humans [37]. To determine whether SP has effects on lipid droplet accumulation in the liver, sections were assessed using ORO staining (Figure 8A).

This showed that liver lipid accumulation in the HFD group was greater than that of the CD group, but that SP treatment significantly reduced the lipid deposition in a dose-dependent fashion. The serum activities of ALT and AST, which are indicative of liver damage (Figure 8B,C), were slightly higher in the control HFD-fed group than in the other groups, including the SP-treated groups, although there were no significant differences. The expression of hepatic SREBP1c, DGAT1, p-AMPK, and CPT1 was also measured in the livers of the mice (Figure 8D,E). SREBP1c and DAGT1 are involved in hepatic lipogenesis [38], and SP treatment markedly reduced their expression. Furthermore, the phosphorylation of AMPK and CPT1 expression were markedly increased by SP treatment. Taken together, these findings suggest that SP reduces lipid accumulation and increases FAO in the liver.

### 3.7. SP Inhibits Lipid Accumulation and Increases the Expression of Markers of Browning in Primary sWAT-Derived Adipocytes

To determine whether SP directly induces browning in sWAT, primary sWAT-derived adipocytes isolated from 5-week-old mice were differentiated in the presence of SP. An MTT assay using 3-(4,5-dimethyl-2-thiazolyl)-2,5-diphenyl-2H-tetrazolium bromide was performed to determine whether SP has any cytotoxic effects, which were not identified at a range of concentrations up to 200 μg/mL (Figure 9A). ORO staining showed that lipid accumulation in primary sWAT-derived adipocytes was significantly inhibited by SP treatment (Figure 9B). SP also markedly reduced lipid accumulation in the primary cells, as indicated that reducing in the expression of the adipogenesis markers C/EBPα and FABP4 (Figure 9C).

In addition, SP induced the expression of CPT1 and UCP1, which are crucial mediators of browning in adipocytes, in a dose-dependent manner (Figure 9D). To compare the effects of SP on the browning program in primary adipocytes with those of a well-characterized inducer cocktail, the cells were incubated with 50 nM T_3_ and 1 µM Rsg, with or without 100 ug/mL of SP. T_3_ and Rsg are known to regulate energy metabolism in adipocytes, and specifically to activate thermogenesis in BAT or beige adipocytes by stimulating β3 adrenergic signaling [39]. According to immunofluorescence analysis (Figure 9E), SP treatment increased mitochondrial activity and the expression of UCP1 in sWAT, and MtRed and UCP1 immunostaining was significantly further upregulated by the addition of SP to the conventional browning inducer cocktail, and there was a synergistic effect of these two treatments on the protein levels of p-AMPK, CPT1, PRDM16, and UCP1 (Figure 9F). This implies that SP treatment reduces lipid accumulation by inhibiting the expression of adipogenic factors and upregulating UCP1.

### 3.8. SP Reduces Lipid Drople Size and Increases UCP1 Expression by Inducing AMPK Phosphorylation

In activating the thermogenic program in adipose tissue, β3 adrenergic signaling upregulates UCP1 expression via an increase in the phosphorylation of AMPK [40]. In addition, AMPK regulates lipid metabolism in adipose tissue, including lipolysis, lipogenesis, and fatty acid metabolism [41]. AMPK acts as a central energy sensor, upregulating thermogenesis by increasing the expression of BAT-specific genes such as PRDM16, PGC1α, and UCP1 [42].

To identify a link between the effects of SP treatment in primary sWAT-derived adipocytes and AMPK activation, we treated adipocytes with 10 μM AICAR or 5 μM dorsomorphin, which are respectively an AMPK activator and an AMPK inhibitor. The number of small lipid droplets was increased by SP and AICAR treatment, as shown in Figure 10A. Furthermore, the western blotting data shown in Figure 10B indicate that SP has a synergistic effect with AICAR to increase p-AMPK and UCP1 expression. By contrast, dorsomorphin treatment markedly inhibited the differentiation of primary cells, reducing the number of lipid droplets. However, the number of small lipid droplets in cells treated with SP and dorsomorphin were slightly increased (Figure 10C). Similarly, the expression of p-AMPK and UCP1 was also lower in dorsomorphin-treated primary cells, whereas SP significantly upregulated these proteins in the adipocytes. Overall, our findings imply that AMPK plays an important role in promoting the browning of WAT in response to SP treatment.

## 4. Discussion

There has been a great deal of recent interest in methods of enhancing energy expenditure in BAT or inducing the transdifferentiation of white to BAT-like adipocytes because these strategies have the potential to prevent the metabolic complications of obesity. The latter mechanism, referred to as “browning”, crucially involves the induction of UCP1 expression in WAT. A number of small molecules, such as berberine, curcumin, and capsaicin, have been shown to activate thermogenic transcriptional factors or regulate key signaling cascades in adipocytes and thus induce browning [43,44,45,46]. In the present study, we aimed to determine whether acid-hydrolyzed SP could reduce lipid accumulation by stimulating energy expenditure, browning, and FAO in WAT via AMPK activation.

Dietary SP is a natural material isolated from *Bombyx mori* that is known to have effects on metabolism [27,47]. In recent studies, it is reported that hydrolyzed proteins are efficiently absorbed in the small intestine [48]. Since SP contains dipeptides and tripeptides including amino acids, dietary SP also good source of nutrients for Intestinal Absorption. It has been shown that silk fibroin inhibits adipocyte differentiation by reducing C/EBPα expression in the 3T3-L1 cell line [49], and a variety of silk proteins are also known to have a protective effect against metabolic diseases [50]. However, the molecular mechanisms of the beneficial effects of SP have yet to be established, and whether it might induce transdifferentiation of WAT is of particular interest. Therefore, we administered SP to HFD-induced obese mice and determined whether this would induce a switch in WAT toward a BAT-like phenotype.

In the present study, we have shown a novel effect of SP to reduce adipogenesis and obesity in HFD-fed mice. As expected, mice fed the HFD became obese and deposited more lipid, including in both the sWAT and vWAT. However, SP treatment markedly reduced the body mass and the adipocyte size of both the sWAT and vWAT of HFD-fed mice. Mature adipocytes in WAT store triglyceride, which is released by lipolysis. We also confirmed that the serum concentrations of triglyceride, total cholesterol, and LDL-cholesterol are higher, and that of HDL-cholesterol is lower in HFD-fed mice, but these defects are ameliorated by SP treatment. Therefore, we also determined the effects of SP on adipogenic genes/proteins and found that the expression of C/EBPα and FABP4 was much lower in SP-treated HFD-fed obese mice and differentiated primary sWAT-derived adipocytes. Thus, the present data suggest that SP can prevent HFD-induced obesity by inhibiting adipogenesis in WAT.

WAT serves as a regulator of a variety of physiologic processes and plays an important role in the regulation of energy balance and lipid homeostasis [51]. In the present study, we have also shown that SP induces the browning of WAT. In vivo and ex vivo experiments showed that SP increased the white adipocyte expression of UCP1, a key marker of browning, as well as that of other vital regulators of this process: PRDM16, PGC1α, and UCP3. The thermogenic capacity of BAT-like adipocytes is conferred by UCP1, which is located in the inner mitochondrial membrane [11], while PGC1α upregulates fatty acid and lipid catabolism, and PRDM16 activates the BAT-selective expression program in white adipocytes [18]. In addition, SP induced mitochondrial biogenesis, as indicated by greater expression of NRF1 and TFAM in both WAT depots. The mitochondria are the site of a number of metabolic pathways in the adipocyte, including β-oxidation, and an enhancement of mitochondrial biogenesis is a crucial feature of WAT browning, because enhancement of mitochondrial biogenesis induces WAT to a BAT-like phenotype [52]. In summary, our data demonstrate that SP induces browning in the WAT of HFD-induced obese mice.

These findings were corroborated by the fact that SP administration also increased the rectal temperature of HFD-fed mice, which probably indicates greater loss of energy as heat. Many researchers have previously demonstrated thermogenesis in animals using indirect thermometers [53,54]. The dissipation of energy as heat is the result of uncoupling in BAT-like adipocytes and involves the transfer of fatty acids from the TGs stored in fat droplets to the mitochondria for FAO [31].

In beige adipocytes, AMPK activation induces the expression of key thermogenic transcription factors, which upregulate the transcription of browning and FAO genes [55]. In SP-treated mice, p-AMPK/AMPK protein levels and CPT1 expression were normalized. FAO is required for the upregulation of UCP1 expression, which is indicative of the phenotypic change from WAT to beige adipocytes [32]. Moreover, mitochondrial biogenesis is accompanied by an upregulation of FAO, and SP treatment increased the expression of the β-oxidation genes *Acox1*, *Cyp4A10*, and *Cyp4a14*. ACOX1 is the first enzyme in the peroxisomal fatty acid β-oxidation pathway, and it has recently been reported that it mediates the increase in energy expenditure and the lean phenotype that results from PPARα activation [56]. Furthermore, CYP4A10 and CYP4A14 are involved in the β- and ω-oxidation of fatty acids in adipose tissue. These genes are induced by PPARα and mediate the conversion of fatty acids to acyl-CoA, which are transported toward the mitochondrial FAO through the action of CPT1 [31]. Thus, our data are consistent with the notion that SP induces the formation of BAT-like adipocytes in WAT by increasing the expression of FAO genes.

Interestingly, SP induced higher expression levels of PRDM16, PGC1α, UCP1, and CPT1 in sWAT than in vWAT. Immunofluorescence analysis also showed that CPT1 and UCP1 expression was much higher in sWAT than in vWAT. Consistent with this, numerous previous mouse studies have shown that sWAT has a larger mitochondrial mass than vWAT and a higher respiratory capacity, attributable to higher expression of mitochondrial respiratory chain components [57,58]. Thus, SP treatment of HFD-fed mice induces browning more efficiently in sWAT than in vWAT.

Lipolysis is accompanied by a potent stimulation of mitochondrial β-oxidation in WAT and is necessary for the browning of WAT. Free fatty acids released by lipolysis can be oxidized in mitochondria [52], and ATGL and HSL are the rate-limiting enzymes in intracellular lipid catabolism. According to a recent report, ATGL-overexpressing mice demonstrate an upregulation of both lipolysis and FAO in WAT, which is accompanied by an upregulation of thermogenesis, resulting in greater energy expenditure [31]. In the present study, SP treatment increased ATGL and HSL expression, implying an upregulation of lipolysis, which would provide substrates for FAO in WAT.

Liver is a key tissue for whole body energy homeostasis but develops non-alcoholic fatty liver disease and specifically steatosis in obese mice and humans [59]. In the present study, hepatic lipid accumulation was ameliorated by SP treatment. Furthermore, SP-treated mice had lower hepatic SREBP1c and DGAT1 expression than HFD-fed control mice, whereas p-AMPK and CPT1 levels were higher in the SP-treated groups. To summarize, SP appears to upregulate FAO and ameliorate fat accumulation in the liver, in addition to improving lipid metabolism in adipose tissue.

We also compared the direct browning effect of the conventional inducers T3 and Rsg and that of SP by measuring the expression levels of p-AMPK, CPT1, and BAT-specific genes, such as PRDM16 and UCP1, in primary sWAT-derived adipocytes. T3 and Rsg are known to regulate thermogenesis, as well as energy balance, by increasing UCP1 expression. In the present study, SP had a synergistic effect with Rsg and T3 to induce the expression of browning-related genes. In addition, differentiated primary cells were assessed to determine whether SP increases mitochondrial biogenesis and UCP1 expression. Immunofluorescence staining for UCP1, and MitoTracker staining to identify mitochondria, were strongly enhanced in cells treated with browning inducers T3 and Rsg. Taking these data together, SP directly stimulates mitochondrial function and upregulates UCP1 expression in white adipocytes.

Finally, we determined whether AMPK phosphorylation is required for the induction of browning and FAO in WAT by SP. AMPK is an essential energy sensor in adipocytes and regulates body mass, liver lipid content, and the thermogenic program in BAT and sWAT [60]. SP treatment of primary adipocytes cultured with AICAR or dorsomorphin caused an increase in AMPK phosphorylation. In addition, the expression level of UCP1 was regulated by both the AMPK inducer and inhibitor in SP-treated cells. Furthermore, the size of lipid droplet in primary cells treated with both AICAR and SP was greatly reduced. These data suggest that AMPK activation induces adipocyte browning by regulating UCP1 expression and fat storage. In conclusion, SP could not only inhibit fat accumulation, but also enhances energy metabolism by regulating UCP1 expression.

## 5. Conclusions

Dietary SP induces WAT-to-BAT transdifferentiation and reduces adiposity by inhibiting adipogenesis and inducing lipolysis. Our data also demonstrate that SP promotes the browning of mouse WAT by increasing the expression of brown adipocyte and FAO genes via AMPK activation. Taking these findings together, we have shown that SP reduces fat accumulation via the AMPK-mediated browning of WAT. Therefore, SP may represent a candidate therapeutic agent for obesity and its metabolic complications.

## Figures and Tables

**Figure 1 nutrients-12-00201-f001:**
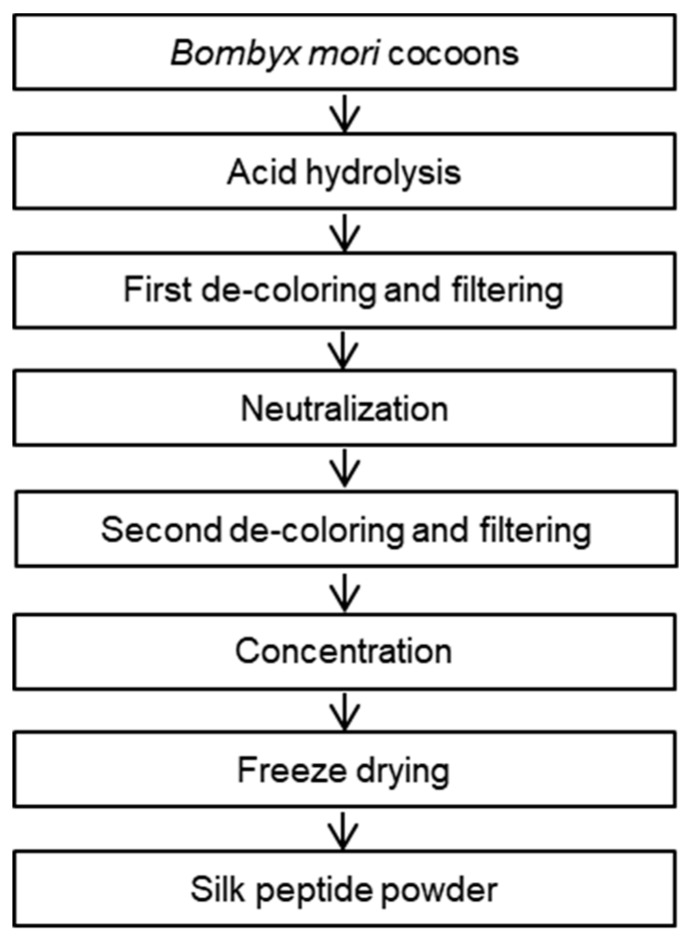
Scheme for the preparation of the silk peptide.

**Figure 2 nutrients-12-00201-f002:**
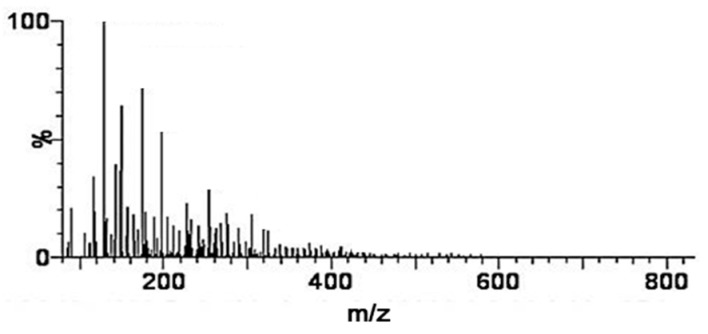
Mass spectrometric analysis of the silk peptide. Typical chromatogram of SP obtained by using high resolution TOF MS ES+ system.

**Figure 3 nutrients-12-00201-f003:**
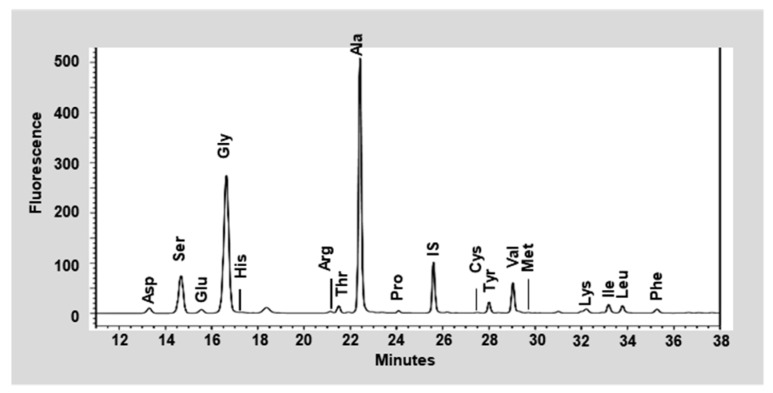
High-performance liquid chromatographic analysis of the silk peptide. HPLC chromatogram for the SP. Abbreviations: Ala, Alanine; Cys, Cysteine; Asp, Aspartic acid; Glu, Glutamic acid; Phe, Phenylalanine; Gly, Glycine; His, Histidine; Ile, Isoleucine; Lys, Lysine; Leu, Leucine; Met, Methionine; Pro, Proline; Arg, Arginine; Ser, Serine; Thr, Threonine; Val, Valine; Tyr, Tyrosine; IS, Interpolation of standard (L-2-aminobutyric acid).

**Figure 4 nutrients-12-00201-f004:**
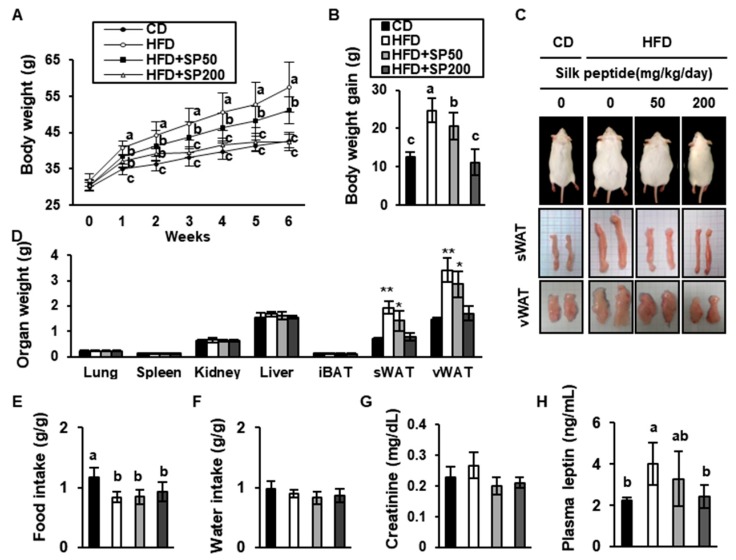
Silk peptide prevents the development of diet-induced obesity in HFD-induced obese mice. (**A**) Weekly body mass measurements, (**B**) body mass gain, and (**C**) photographs of representative mice treated for 6 weeks. (**D**) Effect of SP on organ mass in the CD and HFD groups. * *p* < 0.05 and ** *p* < 0.01 indicates significantly difference from CD group. (**E**) Food intake and (**F**) water consumption per unit body mass. Serum (**G**) creatinine and (**H**) leptin concentrations were measured using colorimetric kits. Data are expressed as mean ± SD (*n* = 8). Values with different letters were significantly different; *p* < 0.05 (a > ab > b > c).

**Figure 5 nutrients-12-00201-f005:**
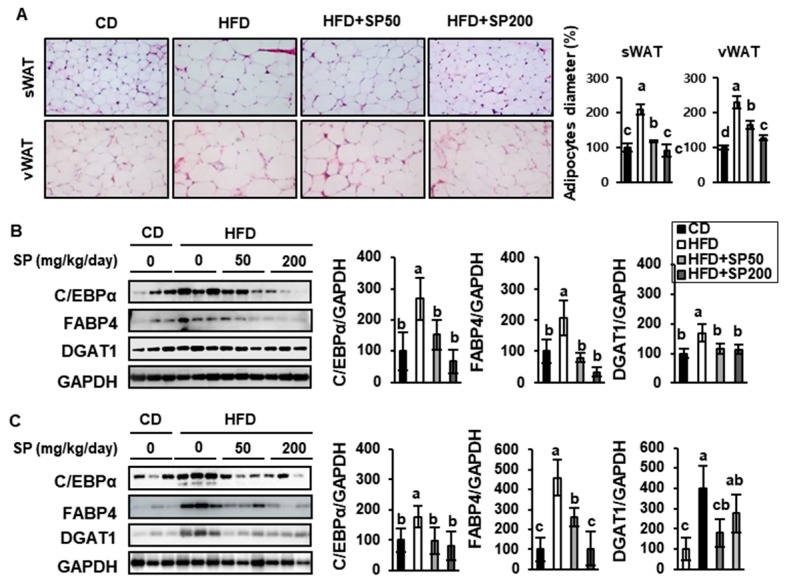
Protective effect of silk peptide against lipid accumulation in white adipose tissue from HFD-induced obese mice. (**A**) Hematoxylin and eosin staining of sWAT and vWAT from mice treated for 6 weeks and quantification of relative adipocyte diameter percentage of sWAT and vWAT (data were collected from H&E-stained sections of mice in each group, 15–30 cells per same area, using Image J software). The expression of adipogenic factors in (**B**) sWAT and (**C**) vWAT were assessed by western blotting. Data are expressed as mean ± SD (*n* = 8). Values with different letters were significantly different; *p* < 0.05 (a > ab > b > bc > c).

**Figure 6 nutrients-12-00201-f006:**
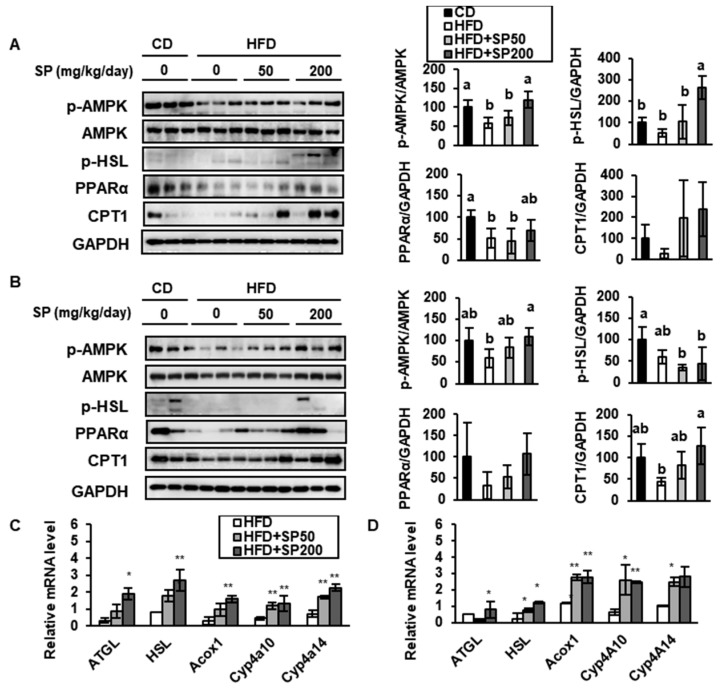
Silk peptide increases fatty acid oxidation in white adipose tissue from HFD-induced obese mice. The expression of p-AMPK, AMPK, p-HSL, PPARα, and CPT1 in (**A**) sWAT and (**B**) vWAT was determined by western blotting. Values with different letters were significantly different; *p* < 0.05 (a > ab > b). The expression of fat metabolism genes in (**C**) sWAT and (**D**) vWAT was measured using qRT-PCR. Data are expressed as mean ± SD (*n* = 8). * *p* < 0.05 and ** *p* < 0.01 indicates significantly difference from HFD group.

**Figure 7 nutrients-12-00201-f007:**
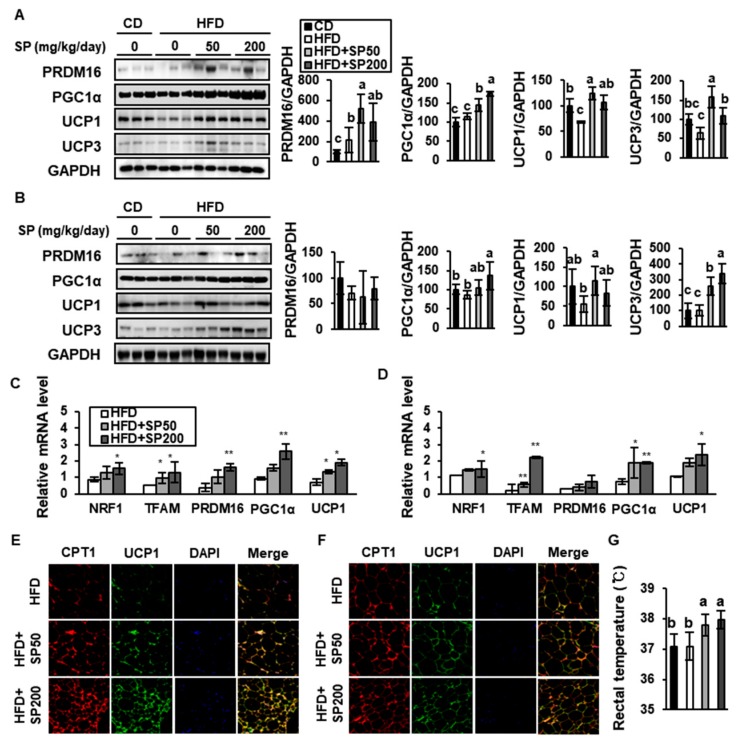
The effect of silk peptide on the browning of white adipose tissue in HFD-induced obese mice. The expression of browning markers (PRDM16, PGC1α, UCP1, and UCP3) in (**A**) sWAT and (**B**) vWAT was determined by western blotting. The expression of thermogenesis-related genes in (**C**) sWAT and (**D**) vWAT was measured using qRT-PCR. * *p* < 0.05 and ** *p* < 0.01 indicates significantly difference from HFD group. The immunofluorescence images of (**E**) sWAT and (**F**) vWAT were captured at 800× magnification. (**G**) Rectal body temperature after 6 weeks of treatment (*n* = 10), measured during the daytime at an ambient temperature of 22 °C. Data are expressed as mean ± SD (*n* = 8). Values with different letters were significantly different; *p* < 0.05 (a > ab > b > bc > c).

**Figure 8 nutrients-12-00201-f008:**
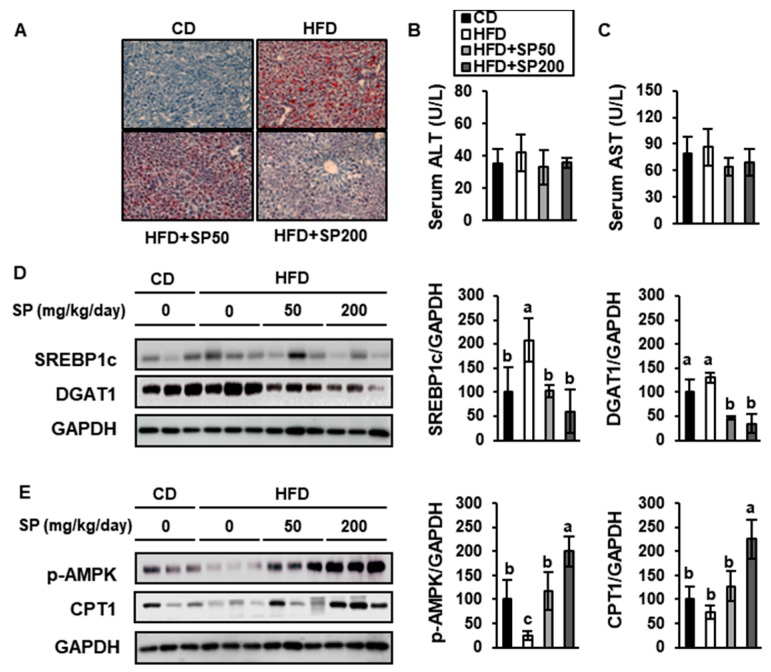
Silk peptide ameliorates hepatic steatosis and induces fatty acid oxidation in HFD-fed mice. (**A**) Representative images of Oil red O staining of liver sections from mice treated for 6 weeks. (**B**) Serum ALT and (**C**) AST were measured using colorimetric kits. The expression levels of proteins involved in (**D**) lipogenesis and (**E**) fatty acid oxidation. Data are expressed as mean ± SD (*n* = 8). Values with different letters were significantly different; *p* < 0.05 (a > b > c).

**Figure 9 nutrients-12-00201-f009:**
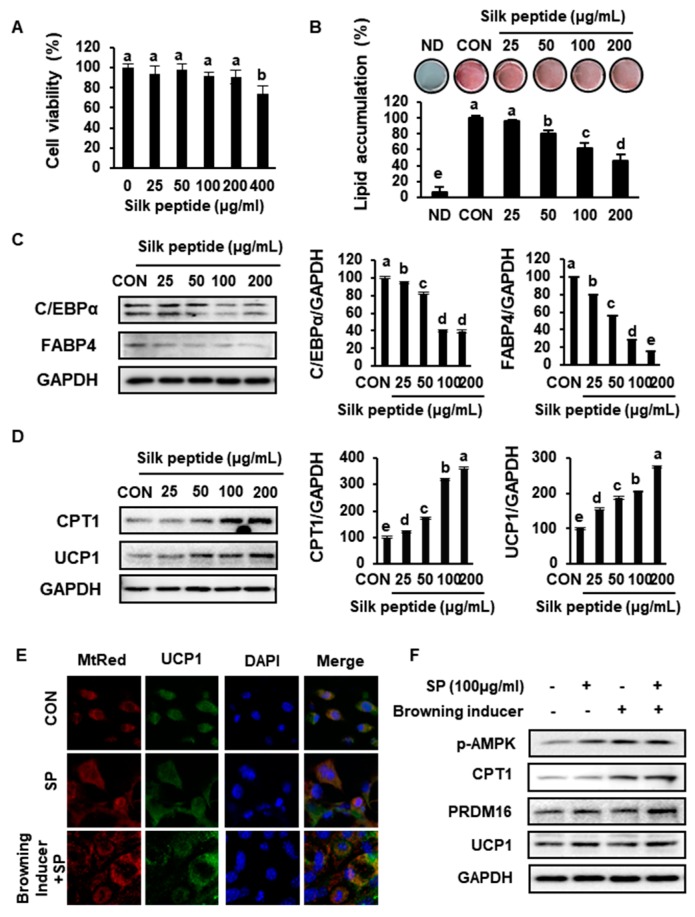
Silk peptide inhibits fat accumulation and increases browning in primary cultures of subcutaneous white adipocytes. (**A**) The viability of primary adipocytes treated with SP for 24 h was assessed using an MTT assay. (**B**) The effect of SP on lipid accumulation was assessed using Oil red O staining. (**C**, **D**) The protein expression levels of C/EBPα, FABP4, CPT1, and UCP1 were measured in subcutaneous white adipocytes. (**E**) Immunofluorescence images of adipocytes were captured at 800× magnification. After differentiation, cells were fixed with methanol and then stained with MitoTracker^®^ 363 Deep Red or anti-UCP1 antibody. (**F**) Browning was promoted in adipocytes cultured with 50 nM T_3_ and 1 mM rosiglitazone and treated with SP, and the protein expression of BAT-specific genes was measured. Data are expressed as mean ± SD (*n* = 4). Values with different letters were significantly different; *p* < 0.05 (a > b > c > d > e).

**Figure 10 nutrients-12-00201-f010:**
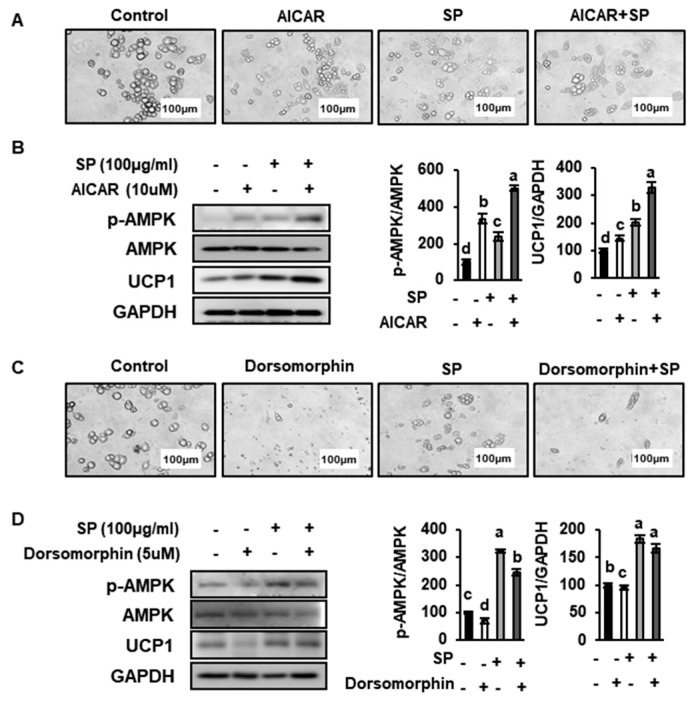
Effects of AMPK on lipid droplet morphology and the expression of UCP1 in primary cultures of subcutaneous white adipocytes. (**A**, **C**) Morphology was evaluated by optical microscopy at 400× magnification. Scale bar: 100 mM. (**B**, **D**) Primary cells were treated with 10 µM AICAR or 5 µM dorsomorphin, and the effects on protein expression were determined using western blotting. Data are expressed as mean ± SD (*n* = 4). Values with different letters were significantly different; *p* < 0.05 (a > b > c > d).

**Table 1 nutrients-12-00201-t001:** Nutrient composition of the silk peptide.

Nutrition Facts	Content (Dry Basis %)
Carbohydrate	6.78
Sugar	0.94
Fat	0.01
Protein	86.80
Sodium	1.79

**Table 2 nutrients-12-00201-t002:** Primer sequences used for quantitative reverse transcription polymerase chain reaction analysis.

Gene	Direction	Sequence (5′–3′)
18s	Forward	GCAATTATTCCCCATGAACG
	Reverse	GGCCTCACTAAACCATCCAA
Acox1	Forward	GCACCTTCGAGGGGGAGAACA
	Reverse	GCGCGAACAAGGTCGACAGAA
ATGL	Forward	TTGGTTCAGTAGGCCATTCC
	Reverse	ACAGTGTCCCCATTCTCAGG
Cyp4a10	Forward	TTCAGAGCCTCCTGGGGGAT
	Reverse	GGAGCAGTGTCAGGGCCACAA
Cyp4a14	Forward	ATGCCTGCCAGATTGCTCACG
	Reverse	GGGTGGGTGGCCAGAGCATAG
HSL	Forward	AGACACCAGCCAACGGATAC
	Reverse	ATCACCCTCGAAGAAGAGCA
NRF1	Forward	TTGGAACAGCAGTGGCAAGA
	Reverse	CTCACTTGCTGATGTATTTACTTCCAT
PGC1α	Forward	ATGTGTCGCCTTCTTGCTCT
	Reverse	ATCTACTGCCTGGGGACCTT
PRDM16	Forward	GATGGGAGATGCTGACGGAT
	Reverse	TGATCTGACACATGGCGAGG
TFAM	Forward	GTCGCATCCCCTCGTCTATC
	Reverse	GCTGGAAAAACACTTCGGAATAC
UCP1	Forward	ACTGCCACACCTCCAGTCAT
	Reverse	CTTTGCCTCACTCAGGATTG

**Table 3 nutrients-12-00201-t003:** Free amino acid composition of the protein component of the silk peptide.

Amino Acid	Content (Dry Basis %)
Glycine	33.06
Alanine	28.09
Serine	11.09
Valine	2.67
Tyrosine	2.46
Aspartic acid	2.45
Glutamic acid	1.78
Threonine	1.22
Cystein	1.04
Isoleucine	0.76
Proline	0.74
Luecine	0.72
Arginine	0.50
Phenylalanine	0.44
Lysine	0.38
Histidine	0.38
Methionine	0.08

**Table 4 nutrients-12-00201-t004:** Effect of silk peptide treatment on blood lipid parameters in HFD-induced obese mice.

Group	Blood Parameter (mg/dL)			
	CD	HFD	HFD	HFD
	Silk Peptide 0 *	Silk Peptide 0 *	Silk Peptide 50 *	Silk Peptide 200 *
Triglycerides	102.8 ± 5.3 ^b^	120.5 ± 6.0 ^a^	97.0 ± 20.0 ^b^	98.3 ± 7.5 ^b^
Total cholesterol	94.5 ± 7.9 ^b^	134.5 ± 5.4 ^a^	87.3 ± 18.0 ^b^	99.3 ± 19.0 ^b^
LDL cholesterol	11.5 ± 1.9 ^b^	14.8 ± 2.5 ^a^	9.0 ± 2.7 ^b^	9.3 ± 2.2 ^b^
HDL cholesterol	120.0 ± 8.2 ^a^	97.8 ± 6.4 ^b^	110.3 ± 13.2 ^a b^	116.3 ± 7.5 ^a^

* (mg/kg/day). Data are expressed as mean ±SD (*n* = 10). Values with different letters are significantly different; *p* < 0.05 (a > b).

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
