# Peer review of "Dietary Silk Peptide Prevents High-Fat Diet-Induced Obesity and Promotes Adipose Browning by Activating AMP-Activated Protein Kinase in Mice"

_nutrients, 2020, doi:10.3390/nu12010201_

Round 1

Reviewer 1 Report

The paper addresses an interesting subject which is the usefulness of silk peptide derived from Bombyx mori cocoons. The authors investigated the effects of dietary silk peptide in high-fat diet-fed obese mice. In vitro experiments had been also performed.

The study is well designed according to the planned objectives. The paper is generally well written and well organized.

A few issues, however, need to be addressed:

In the Material and methods section please consider changing the order of the tables. In my opinion table, 2 on page 3 should be numbered as table 1,  and table 3 should be table 2, etc. Please provide additional information about the method of analysis of qPCR data. Was it the 2-ΔΔCT method ? In table 4 please add units after each blood parameter in the left column. Please consider type in ”Study group” instead of “Blood parameter (mg/ml)”  in the first row of table 4.

Author Response

Dear Chief in Editor and reviewers,

Thank you for considering our manuscript for publication in Nutrients.

We are very pleasure to have been given the opportunity to revise our manuscript. We have addressed the reviewer’s comments point-by-point and made the necessary changes to the manuscript.

We hope that the manuscript is now acceptable for publication in Nutrients.

Thank you again for your hard work, it is greatly appreciated.

Sincerely,

Boo-Yong Lee

1)In the Material and methods section please consider changing the order of the tables. In my opinion table, 2 on page 3 should be numbered as table 1,  and table 3 should be table 2, etc.

→ Response: We changed the Table numbering 1-4 as indicated. Thank you for your pointing out our mistake.

2) Please provide additional information about the method of analysis of qPCR data. Was it the 2-ΔΔCT method ?

→ Response: As reviewer’s comment, as add additional information in Method 2.8.: The mRNA levels were calculated as a ratio, using the 2−ΔΔCT method by using Bio-Rad software (Quantity One 4.62; Bio-Rad), for comparing the relative mRNA expression levels between different groups in the qPCR.

3) In table 4 please add units after each blood parameter in the left column. Please consider type in ”Study group” instead of “Blood parameter (mg/ml)”  in the first row of table 4.

→ Response: Actually we analyzed the blood parameters as unit mg/dL (Creatinine, Triglyceride, Total cholesterol, LDL cholesterol, and HDL cholesterol). Thank you for your pointing out our mistake.

Reviewer 2 Report

Authors here studied the mechanism whereby silk peptide (SP), obtained by acid hydrolysis of cocoons of Bombyx mori, prevents high-fat diet-induced obesity. In this manuscript, they report that SP has a browning effect in white adipose tissue by upregulating phosphorylation of AMP-activated protein kinase and expression of uncoupling protein 1, that SP suppresses adipogenesis and promotes fatty acid oxidation, and that SP has potential to be used as an anti-obesity drug.

Before they studied the anti-obesity effect of SP, they characterized SP. The methods used are described on p. 2; however, the description seems to be unsatisfactory for the following reasons:
1) The preparation of SP is briefly described on lines 65-68; however, more information (e.g., conditions for acid hydrolysis, filtering and desalting methods used) would be required.
2) It is written that "The nutrient composition of the SP obtained was analyzed in duplicate (lines 68-69)"; however, the methods used are not described.
3) The results of the nutrient composition analysis are described on line 69, citing Table 2; they may be described in the Results section. In addition, Tables 2 and 3 on p. 3 appear before Table 1, which appears on p. 7.
4) What is "the reaction mixture" described on line 71?
5) On line 75, it is written that "the results are presented in Figure 2". Here, "the results" seem to be HPLC data on "the reaction mixture"; however, Figure 2 shows data obtained by mass spectrometric analysis. The description thus seems to be confused. In addition, the results may be described in the Results section.
6) The results obtained by amino acid analysis of the free amino acid composition of SP is described on line 78. The results may be described in the Results section.

The results on characterization of SP are described on pp. 6-7 (lines 178-201).
There are several questions/comments:
1) On line 178, there is a description, "acid-hydrolyzed silk peptide". It is likely that "acid-hydrolyzed silk peptide" usually means a mixture of products obtained by acid hydrolysis of silk peptide; however, "acid-hydrolyzed silk peptide" seems to be used here to mean silk peptide itself. If "acid-hydrolyzed silk peptide (SP)" means silk peptide (SP) itself, the use of "acid-hydrolyzed" must be avoided. The same applies to "acid-hydrolyzed" in other sentences.
2) It is written that "The mean molecular weights of SP were measured by ... (on p. 2, lines 69-70)", and that "SP was prepared as a 400 mM stock solution in distilled water ... (p. 4, line 96)". However, it is not reported what value was used as the mean molecular weight of SP. This value must be provided here.
3) The data shown in Figure 2 suggest that SP contains dipeptides and tripeptides, which are reported to be effectively absorbed in the intestine. It is preferable to provide information on adsorption of SP or discuss it in the Discussion section.
4) In the legend to Figure 2, it is written that "Data shows a tandem mass spectrum ... (p. 6, lines 194-195)"; however, Figure 2 does not seem to show a tandem mass spectrum.
5) It is written that "Further quantitative analysis of the SP preparation was performed by HPLC and the data are presented in Figure 3 (p. 6, lines 183-185)". However, it is likely that Figure 3 shows data obtained by amino acid analysis of the acid hydrolysate of SP.

The descriptions in "2.1 Preparation of SP from Bombyx mori (p. 2)" and in "3.1. Nutritional analysis of the acid-hydrolyzed silk peptide (p. 6)" must be confirmed and revised.

I have no serious comments on the reported effects of SP.

During the review, I noticed several minor points to be considered by the authors. These are listed below (suggestions are indicated by "-->").

p. 1, line 20:
AMPK
--> AMP-activated protein kinase (AMPK)

p. 1, line 20:
UCP1
--> uncoupling protein 1 (UCP1)

p. 1, line 43:
, that is
--> , which is

p. 4, line 100:
were added to
--> was added to

p. 4, line 113:
intake, and water consumption of the mice was
--> intake, and water consumption of the mice were

p. 4, line 116:
2.3.
--> 2.4.
Comment: Subsequent section numbers should be corrected.

p. 4, lines 122-123:
Commercial enzyme-linked immunosorbent assay (ELISA) kits (Abcam and Biocompare, CA, USA)
--> Commercial enzyme-linked immunosorbent assay (ELISA)/calorimetric assay kits (Abcam and Biocompare, CA, USA)

p. 4, line 127:
measured at 570 nm
--> measured at appropriate wavelengths

p. 4, line 129:
vWAT
--> visceral WAT (vWAT)

p. 5, line 136-137:
Fluorescein isothiocyanate (FITC)-conjugated (dilution, 1:500) and Alexa Fluor™ 594-conjugated secondary antibodies
--> Fluorescein isothiocyanate (FITC)-conjugated (dilution, 1:500) and Alexa Fluor™ 594-conjugated (dilution, 1:???) secondary antibodies

p. 5, line 145:
at temperature
-->? at room temperature

p. 5, line 151:
Cells or tissue were
--> Cells or tissue was

p. 5, line 156:
(p-HSL, Ser563)
--> (p-HSL, Ser563),

p. 5, lines 159-160:
peroxisome proliferator-activated receptor PPARα
--> peroxisome proliferator-activated receptor alpha (PPARα)

p. 5, lines 150-161:
Comment: Incubation with secondary antibodies is not described. The method used for detection of immunostained bands is not described, either.

p. 6, line 175:
mRNA data data
--> mRNA data data

p. 7, line 209:
visceral WAT (vWAT)
--> vWAT
Comment: "vWAT" was introduced on p. 4 (line 129).

p. 8, line 224:
in the HFD than
--> in the HFD group than

p. 8, line 226:
in the HFD than
--> in the HFD group than

p. 8, line 232:
uces WAT depot size
-->? SP reduces WAT depot size
Comment: Some words/letters are dropped.

p. 9, lines 263-265:
HSL and ATGL are ... and can induce browning in WAT.
Comment: Reference citation is preferable.

p. 9, lines 269-270:
... Acox1, Cyp4a10, and Cyp4a14, which are required for peroxisomal and mitochondrial FAO.
Comment: Reference citation is preferable.

p. 11, Figure 7c:
PGC1
--> PGC1a

p. 15, lines 410-412:
the serum concentrations of triglyceride, total cholesterol, and LDL-cholesterol are lower, and that of HDL-cholesterol is higher in HFD-fed mice,
--> the serum concentrations of triglyceride, total cholesterol, and LDL-cholesterol are higher, and that of HDL-cholesterol is lower in HFD-fed mice,

p. 15, line 438:
CTP1
--> CPT1

p. 15, line 450:
immunofluorescence also
--> immunofluorescence analysis also

p. 15, lines 484-485:
Furthermore, lipid droplet accumulation in primary cells treated with both AICAR and SP was greatly reduced.
Comment: Is this description correct?

p. 15, line 489:
Discussion
--> Conclusions

Author Response

Dear Chief in Editor and reviewers,

Thank you for considering our manuscript for publication in Nutrients.

We are very pleasure to have been given the opportunity to revise our manuscript. We have addressed the reviewer’s comments point-by-point and made the necessary changes to the manuscript.

We hope that the manuscript is now acceptable for publication in Nutrients.

Thank you again for your hard work, it is greatly appreciated.

Sincerely,

Boo-Yong Lee

<The methods used are described on p. 2>

1) The preparation of SP is briefly described on lines 65-68; however, more information (e.g., conditions for acid hydrolysis, filtering and desalting methods used) would be required.

→ Response: We appreciate your suggestion. However, preparation information of SP is confidential and may constitute legally privileged information of Worldway Co., Ltd. Thank you for understanding that we could not add manufacture process information.

2) It is written that "The nutrient composition of the SP obtained was analyzed in duplicate (lines 68-69)"; however, the methods used are not described.

→ Response: The nutrient composition of the SP analyzed and estimated by International official methods of analysis (AOAC) methods. In detail, carbohydrates and sugars in an activated charcoal column and a later elution with different proportions of ethanol (AOAC 954.11) to fractionate them selectively according to their degree of polymerization. Crude fat of SP were extracted using the soxhlet apparatus with hexane for 4 hours (AOAC 920.153) and determined gravimetrically. Crude protein was estimated by AOAC 968.06 method, through an acid digestion and nitrogen distillation using Kjeldahl method. Lastly, sodium content in SP was conducted following the AOAC 984.27 method. We added this information in Method 2.1.

3) The results of the nutrient composition analysis are described on line 69, citing Table 2; they may be described in the Results section. In addition, Tables 2 and 3 on p. 3 appear before Table 1, which appears on p. 7.

→ Response: We changed the Table numbering 1-4 as indicated. Thank you for your pointing out our mistake.

4) What is "the reaction mixture" described on line 71?

→ Response: We used the SP sample for HPLC analysis, which means that SP dissolved in 10mM Sodium phosphate buffer with methanol (4:1)”. We rewrote the sentence.

5) On line 75, it is written that "the results are presented in Figure 2". Here, "the results" seem to be HPLC data on "the reaction mixture"; however, Figure 2 shows data obtained by mass spectrometric analysis. The description thus seems to be confused. In addition, the results may be described in the Results section.

→ Response: In regard with answer of question 4), our mass spectrometric analysis data was indicates various molecular weights of SP. Therefore, we could suggest that “the mean molecular weight of the components of the SP ranged from 150 to 300 Da and that of the acid-hydrolyzed SP was <500 Da” in Result 3.1.

6) The results obtained by amino acid analysis of the free amino acid composition of SP is described on line 78. The results may be described in the Results section.

→ Response: As your kind recommend, we deleted “and the results of which are shown in Table” on line 78, and described the results in Result 3.1.

<The results on characterization of SP are described on pp. 6-7 (lines178-201)>

7) On line 178, there is a description, "acid-hydrolyzed silk peptide". It is likely that "acid-hydrolyzed silk peptide" usually means a mixture of products obtained by acid hydrolysis of silk peptide; however, "acid-hydrolyzed silk peptide" seems to be used here to mean silk peptide itself. If "acid-hydrolyzed silk peptide (SP)" means silk peptide (SP) itself, the use of "acid-hydrolyzed" must be avoided. The same applies to "acid-hydrolyzed" in other sentences.

→ Response: Thank you for your suggestion. Acid-hydrolyzed silk peptide SP is same as SP in our study. Thereby, we revised “acid-hydrolyzed silk peptide (SP)” to “silk peptide (SP)” to unify the term in draft.

8) It is written that "The mean molecular weights of SP were measured by ... (on p. 2, lines 69-70)", and that "SP was prepared as a 400 mM stock solution in distilled water ... (p. 4, line 96)". However, it is not reported what value was used as the mean molecular weight of SP. This value must be provided here.

→ Response: We answered in 5) about mass spectrometric analysis. 400 mM stock solution of SP was prepared in distilled water and used for treatment in primary cell culture, since solvents has toxicity on cell.

9) The data shown in Figure 2 suggest that SP contains dipeptides and tripeptides, which are reported to be effectively absorbed in the intestine. It is preferable to provide information on adsorption of SP or discuss it in the Discussion section.

→ Response: As your kind comment, we add these sentences in Discussion section on line 408, as below. “In recent studies, it is reported that hydrolyzed proteins are efficiently absorbed in the small intestine. Since SP contains dipeptides and tripeptides including amino acids, dietary SP also good source of nutrients for Intestinal Absorption.”[Reference 49];Kiela, P.R.; Ghishan, F.K. Physiology of Intestinal Absorption and Secretion. Best practice & research. Clinical gastroenterology 2016, 30, 145-159, doi:10.1016/j.bpg.2016.02.007.

10)In the legend to Figure 2, it is written that "Data shows a tandem mass spectrum ... (p. 6, lines 194-195)"; however, Figure 2 does not seem to show a tandem mass spectrum.

→ Response: We deleted this sentence.

11) It is written that "Further quantitative analysis of the SP preparation was performed by HPLC and the data are presented in Figure 3 (p. 6, lines 183-185)". However, it is likely that Figure 3 shows data obtained by amino acid analysis of the acid hydrolysate of SP.

→ Response: As we answered in 7), the acid-hydrolyzed silk peptide SP we used is same as SP in our study. Thereby, Figure 3 indicated HPCL data of SP.

12) The descriptions in "2.1 Preparation of SP from Bombyx mori (p. 2)" and in "3.1. Nutritional analysis of the acid-hydrolyzed silk peptide (p. 6)" must be confirmed and revised.

→ Response: We confirmed and revised the nutritional analysis information in detailed in Method 2.1 section. Thank you for as your comment.

13) I have no serious comments on the reported effects of SP.

→ Response: We appreciate your revision.

14) During the review, I noticed several minor points to be considered by the authors. These are listed below (suggestions are indicated by "-->").

p. 1, line 20:AMPK--> AMP-activated protein kinase (AMPK)

→ Response: We revised as your comment.

p. 1, line 20:UCP1--> uncoupling protein 1 (UCP1)

→ Response: We revised as your comment.

p. 1, line 43:, that is--> , which is

→ Response: We revised as your comment.

p. 4, line 100:were added to--> was added to

→ Response: We revised as your comment.

p. 4, line 113:intake, and water consumption of the mice was--> intake, and water consumption of the mice were

→ Response: We revised as your comment.

p. 4, line 116:2.3.--> 2.4.Comment: Subsequent section numbers should be corrected.

→ Response: We changed the numbering 2.1-2.10 as indicated.

p. 4, lines 122-123:Commercial enzyme-linked immunosorbent assay (ELISA) kits (Abcam and Biocompare, CA, USA)--> Commercial enzyme-linked immunosorbent assay (ELISA)/calorimetric assay kits (Abcam and Biocompare, CA, USA)

→ Response: We revised as your comment.

p. 4, line 127:measured at 570 nm--> measured at appropriate wavelengths

→ Response: We revised as your comment.

p. 4, line 129:vWAT--> visceral WAT (vWAT)

→ Response: We revised as your comment.

p. 5, line 136-137:Fluorescein isothiocyanate (FITC)-conjugated (dilution, 1:500) and Alexa Fluor™ 594-conjugated secondary antibodies--> Fluorescein isothiocyanate (FITC)-conjugated (dilution, 1:500) and Alexa Fluor™ 594-conjugated (dilution, 1:???) secondary antibodies

→ Response: We confirmed and revised as “Fluorescein isothiocyanate (FITC)-conjugated (dilution, 1:1000) and Alexa Fluor™ 594-conjugated (dilution, 1:1000) secondary antibodies were used.”

p. 5, line 145:at temperature-->? at room temperature

→ Response: We revised as your comment.

p. 5, line 151:Cells or tissue were--> Cells or tissue was

→ Response: We revised as your comment.

p. 5, line 156:(p-HSL, Ser563)--> (p-HSL, Ser563),

→ Response: We revised as your comment.

p. 5, lines 159-160:peroxisome proliferator-activated receptor PPARα--> peroxisome proliferator-activated receptor alpha (PPARα)

→ Response: We revised as your comment.

p. 5, lines 150-161:Comment: Incubation with secondary antibodies is not described. The method used for detection of immunostained bands is not described, either.

→ Response: We confirmed and revised as “After washing, the membranes were incubated for 4 h with secondary antibodies conjugated with horseradish peroxidase (1:1000, Santa Cruz Biotechnology) in 5% non-fat dried milk. Reactive band was obtained by chemiluminescence and LAS image software (Fuji, NY, USA).”

p. 6, line 175:mRNA data data--> mRNA data data

→ Response: We deleted as your comment.

p. 7, line 209:visceral WAT (vWAT)--> vWATComment: "vWAT" was introduced on p. 4 (line 129).

→ Response: We revised as your comment.

p. 8, line 224:in the HFD than--> in the HFD group than

→ Response: We revised as your comment.

p. 8, line 226:in the HFD than--> in the HFD group than

→ Response: We revised as your comment.

p. 8, line 232:uces WAT depot size-->? SP reduces WAT depot size Comment: Some words/letters are dropped.

→ Response: We confirmed and revised as “3.2. SP reduces WAT depot size and downregulates adipogenic gene expression”

p. 9, lines 263-265:HSL and ATGL are ... and can induce browning in WAT.Comment: Reference citation is preferable.

→ Response: We add a new reference as [34]; Morak, M.; Schmidinger, H.; Riesenhuber, G.; Rechberger, G.N.; Kollroser, M.; Haemmerle, G.; Zechner, R.; Kronenberg, F.; Hermetter, A. Adipose triglyceride lipase (ATGL) and hormone-sensitive lipase (HSL) deficiencies affect expression of lipolytic activities in mouse adipose tissues. Molecular & cellular proteomics 2012, 11, 1777-1789.

p. 9, lines 269-270:... Acox1, Cyp4a10, and Cyp4a14, which are required for peroxisomal and mitochondrial FAO.Comment: Reference citation is preferable.

→ Response: Response: We add a reference as [28].

p. 11, Figure 7c:PGC1--> PGC1a

→ Response: We revised as your comment.

p. 15, lines 410-412:the serum concentrations of triglyceride, total cholesterol, and LDL-cholesterol are lower, and that of HDL-cholesterol is higher in HFD-fed mice,--> the serum concentrations of triglyceride, total cholesterol, and LDL-cholesterol are higher, and that of HDL-cholesterol is lower in HFD-fed mice,

→ Response: We revised as your comment.

p. 15, line 438:CTP1--> CPT1

→ Response: We revised as your comment.

p. 15, line 450:immunofluorescence also--> immunofluorescence analysis also

→ Response: We revised as your comment.

p. 15, lines 484-485:Furthermore, lipid droplet accumulation in primary cells treated with both AICAR and SP was greatly reduced.Comment: Is this description correct?

→ Response: We revised as your comment as “the size of lipid droplet in primary cells treated with both AICAR and SP was greatly reduced”.

p. 15, line 489:Discussion--> Conclusions

→ Response: We revised as your comment.
